# FAT-TO-THIN POLICY OPTIMIZATION: OFFLINE RL WITH SPARSE POLICIES

**Lingwei Zhu**[*]
University of Tokyo
lingwei4@ualberta.ca

**Han Wang**[*]
University of Alberta
han8@ualberta.ca

**Yukie Nagai**
University of Tokyo

## ABSTRACT

Sparse continuous policies are distributions that can choose some actions at random yet keep strictly zero probability for the other actions, which are radically different from the Gaussian. They have important real-world implications, e.g. in modeling safety-critical tasks like medicine. The combination of offline reinforcement learning and sparse policies provides a novel paradigm that enables learning completely from logged datasets a safety-aware sparse policy. However, sparse policies can cause difficulty with the existing offline algorithms which require evaluating actions that fall outside of the current support. In this paper, we propose the first offline policy optimization algorithm that tackles this challenge: Fat-to-Thin Policy Optimization (FtTPO). Specifically, we maintain a fat (heavy-tailed) proposal policy that effectively learns from the dataset and injects knowledge to a thin (sparse) policy, which is responsible for interacting with the environment. We instantiate FtTPO with the general $q$-Gaussian family that encompasses both heavy-tailed and sparse policies and verify that it performs favorably in a safety-critical treatment simulation and the standard MuJoCo suite. Our code is available at https://github.com/lingweizhu/fat2thin.

## 1 INTRODUCTION

Sparse continuous policies are distributions that can choose some actions at random, yet can maintain strictly zero probability for the other actions. Therefore, they present a radically different solution than the Gaussian policy which has been standard in the existing policy optimization algorithms (Haarnoja et al., 2018; Abdolmaleki et al., 2018). Sparse policies have important real-world implications, e.g. in safety-critical tasks such as treatment where dangerous actions should never be picked but some other actions should be explored (Fatemi et al., 2021; Yu et al., 2021). Infinite-support policies like the Gaussian or the heavy-tailed distributions (Kobayashi, 2019; Zhu et al., 2024b) are hence not suitable in this regard. When sparse policies coupled with offline reinforcement learning (RL), an attractive and novel paradigm emerges: a safe sparse policy can be learned completely from a logged dataset without potentially harming the environment in an online manner.

However, sparse policies pose a challenge to the existing offline deep RL algorithms: the dataset actions can fall outside of the sparse policy's support, leading to undefined log-likelihood and hence learning failure. The same issue persists for off-policy algorithms since offline learning can be seen as an extreme case of off-policy learning. Note that this issue does not occur for infinite-support policies like the Gaussian but rather is inherent to all sparse policies. To the best of our knowledge, there is no systematic solution to the problem of out-of-support actions in offline learning incurred by sparse policies. Existing methods resort to ad hoc solutions such as approximating the sparse policy with the Gaussian (Lee et al., 2020; Xu et al., 2023) or replacing out-of-support actions with random in-support actions (Zhu et al., 2024b).

In this paper we propose Fat-to-Thin Policy Optimization (FtTPO) that for the first time addresses the problem of offline learning with out-of-support actions induced by sparse policies. Our method consists of two steps: learning an infinite-support policy (either the Gaussian or the heavy-tailed) from the dataset, then imparting its knowledge to a sparse policy. Since there is no prior work

---

[*]indicates joint first authors.

directly comparable, we compare FtTPO against the existing ad hoc tricks and popular offline algorithms. As it is commonly perceived that the sparse policies are inherently handicapped at the exploration-exploitation tradeoff, we find it surprising that the sparse policy learned by FtTPO can outperform full-support policies: FtTPO competes favorably against the popular offline algorithms that use the Gaussian by default. To summarize, our contributions include: (1) we are the first to investigate the out-of-support action issue with offline learning incurred by sparse policies; (2) we propose Fat-to-Thin Policy Optimization: the first deep offline RL framework for learning sparse policies; (3) we verify the sparse policy learned by FtTPO can indeed concentrate on a small band of actions. Therefore, it outperforms existing popular baselines on a safety-critical treatment simulated environment and the MuJoCo suite.

## 2 BACKGROUND

We focus on discounted Markov Decision Processes (MDPs) expressed by the tuple $(\mathcal{S}, \mathcal{A}, P, r, \gamma)$, where $\mathcal{S}$ and $\mathcal{A}$ denote state space and action space, respectively. Let $\Delta(\mathcal{X})$ denote the set of probability distributions over $\mathcal{X}$. $P : \mathcal{S} \times \mathcal{A} \to \Delta(\mathcal{S})$ denotes the transition probability function, and $r(s, a)$ defines the reward associated with that transition. $\gamma \in (0, 1)$ is the discount factor. A policy $\pi : \mathcal{S} \to \Delta(\mathcal{A})$ is a mapping from the state space to distributions over actions. In this paper we focus on the offline setting where we learn from a fixed dataset $\mathcal{D}$ that stores transitions. We denote the behavior policy that generates the dataset by $\pi_{\mathcal{D}}$. The learninig goal is to search for an optimal policy that maximizes long-term accumulated rewards. We define the action value and state value as $Q^\pi(s, a) = \mathbb{E}_\pi \left[ \sum_{t=0}^\infty \gamma^t r(s_t, a_t) | s_0 = s, a_0 = a \right], V^\pi(s) = \mathbb{E}_\pi \left[ Q^\pi(s, a) \right].$

We specifically consider policies defined by the deformed $q$-exponential function. The $q$-exponential function and its unique inverse function $q$-logarithm are defined by (Naudts, 2002):

$$\exp_q x = \begin{cases} \exp x, & q = 1 \\ [1 + (1 - q)x]_+^{\frac{1}{1-q}}, & q \neq 1 \end{cases} \qquad \ln_q x := \begin{cases} \ln x, & q = 1 \\ \frac{x^{1-q}-1}{1-q}, & q \neq 1, \end{cases}$$

where $[\cdot]_+ = \max\{\cdot, 0\}$ sets negative part of the input to zero. Note that $\exp_q xy \neq \exp_q x \exp_q y$ unless $q = 1$ and the same for $\ln_q xy$ (Tsallis, 2009). When $q < 1$, it is clear that for the $q$-exp truncates $x < -\frac{1}{1-q}$, i.e.

$$\exp_{q<1} x = \mathbb{1}\{(1 + (1 - q)x)^{\frac{1}{1-q}} \geq 0\} \cdot (1 + (1 - q)x)^{\frac{1}{1-q}}.$$

We can use this property to define sparse distributions. Prior works focused on discrete sparsemax policies (Martins & Astudillo, 2016; Lee et al., 2018; Chow et al., 2018; Lee et al., 2020). In this paper we consider continuous sparse policies, especially the $q$-Gaussian (Naudts, 2010; Furuichi, 2010; Matsuzoe & Ohara, 2011).

## 3 OFFLINE LEARNING WITH SPARSE POLICIES

Offline RL algorithms typically require evaluating actions produced by behavior policies using the current policy, e.g. in computing the log-likelihood. The Gaussian policy as the standard choice does not incur any issue since it is an infinite-support distribution that can always yield nonzero probability for offline actions. By contrast, the actions may fall outside the support of a sparse policy, resulting in numerical issues or even failed learning.

### 3.1 WHEN SPARSE POLICIES CAN FAIL

For simplicity, we illustrate the sparse policy issue by assuming an actor-critic framework where the actor minimizes the forward KL divergence between the desired policy and the parametrized policy. This setting is common in offline RL and has several popular variants (Jaques et al., 2020; Wu et al., 2020; Nair et al., 2021). The loss can be written as:

$$\begin{aligned} \mathcal{L}_{\text{ForwardKL}}(\phi) : &= \mathbb{E}_{s \sim \mathcal{D}} \left[ D_{KL}(\pi_{\text{desired}}(\cdot|s) \,||\, \pi_\phi(\cdot|s)) \right] \\ &= \mathbb{E}_{\substack{s \sim \mathcal{D} \\ a \sim \pi_{\text{desired}}}} \left[ \ln \pi_{\text{desired}}(a|s) - \ln \pi_\phi(a|s) \right], \\ &= \mathbb{E}_{\substack{s \sim \mathcal{D} \\ a \sim \pi_{\text{desired}}}} \left[ - \ln \pi_\phi(a|s) \right]. \end{aligned} \qquad (1)$$

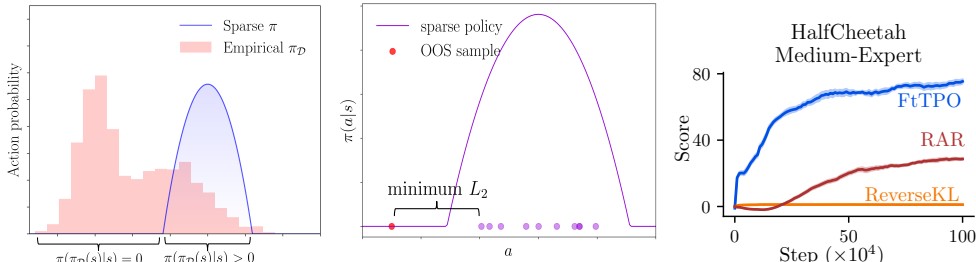

Figure 1: (Left) Illustration of offline learning with a sparse policy. The red histogram represents an empirical behavior policy. The blue distribution indicates the sparse policy being learned. Actions outside the intersection have zero probability under the learned policy. Therefore, their log-likelihood will return $-\infty$. (Middle) The random action replacement (RAR) trick used by (Zhu et al., 2024b). When an out-of-support (OOS) action appears as the red dot, it is replaced by the nearest in-support action (purple dots) sampled from the policy in the $L_2$ sense. This method falls short in high dimensional spaces like the MuJoCo suite. (Right) Performance of the proposed FtTPO against the ad hoc tricks RAR and reverse KL on the MuJoCo Halfcheetah environment, averaged over 10 seeds.

Minimizing this loss amounts to maximizing the log-likelihood of $\pi_\phi$ under the actions sampled from the desired policy. The last line ignores the term not depending on the optimization variable $\phi$. Typically, $\pi_{\text{desired}}$ is set to the behavior policy $\pi_\mathcal{D}$ (or sampling only from the dataset for in-sample learning (Fujimoto et al., 2019)). It is clear that when $\pi_\phi$ is specified as an infinite-support policy like the Gaussian, optimizing $\ln \pi_\phi(\pi_\mathcal{D}(s)|s)$ does not incur any issue. However, when $\pi_\phi$ is designated to be a sparse policy, its support can be disjoint with the behavior policy, i.e. $\pi_\phi(\pi_\mathcal{D}(s)|s) = 0$ and therefore $\ln \pi_\phi(\pi_\mathcal{D}(s)|s)$ returns $-\infty$. Figure 1 left illustrates this process. The red histogram represents an empirical behavior policy. The blue policy indicates the sparse policy being learned. Behavior actions outside the intersection cannot be used in log-probability evaluation.

## 3.2 Ad Hoc Solutions

Since the issue of out-of-support actions have not been studied before, there is no systematic solution to the best of our knowledge. We discuss some ad hoc tricks used by existing works. These tricks include random action replacement or reverse KL loss minimization (Zhu et al., 2024b). Random action replacement (RAR) refers to replacing the out-of-support action with the nearest in-support action sampled from the current policy, as visualized in the middle of Figure 1. However, it should be noted that this method becomes increasingly ineffective as dimension grows, since in high dimension spaces samples are increasingly concentrated inside the ellipsoid (Martins et al., 2022). As a result, it becomes more and more difficult to sample actions near the boundary of the sparse policy. In Figure 1 we visualize the performance of RAR which shows slow learning on the example environment.

Another potential solution is to avoid offline actions from the behavior policy. For example, this can be done by sampling from the learning policy itself. By reversing the direction of the KL divergence in Eq. (1) the loss becomes a reverse KL divergence (Chan et al., 2022):

$$
\begin{aligned}
\mathcal{L}_{\text{ReverseKL}}(\phi) : &= \mathbb{E}_{s\sim\mathcal{D}}\left[D_{KL}(\pi_\phi(\cdot|s)\,||\,\pi_\mathcal{D}(\cdot|s))\right] \\
&= \mathbb{E}_{\substack{s\sim\mathcal{D}\\a\sim\pi_\phi}}\left[\ln \pi_\phi(a|s) - \ln \pi_\mathcal{D}(a|s)\right].
\end{aligned}
\tag{2}
$$

Since actions are now sampled from $\pi_\phi$, no out-of-support issue will occur. However, in practice we find this method performs poorly, as can be seen from Figure 1. We conjecture that the inability of reverse KL alone is due to the finite support of sparse policies, which cause the sampled actions to concentrate around its mode and therefore no significant update can be performed, resulting in extremely slow learning.

## 4 Fat-to-Thin Policy Optimization

In Section 3 we discussed several important considerations underlying popular offline learning methods. They can be instructive even for learning sparse policies, e.g. using samples from the

dataset directly or from a learned behavior policy for learning stability. Our proposed method builds upon these considerations and takes inspiration from recent advances in two-stage actor-critic methods (Neumann et al., 2023). We discuss related work in more detail in Section 6.

## 4.1 TWO-STAGE LEARNING

Recent studies explore two-stage actor-critic methods that aim to efficiently learn an unbiased policy. The first policy, called the **proposal policy**, often maximizes biased reward (e.g. with entropy bonus). It is used to generate actions for the second stage. The second policy – called the **actor policy** – maximizes unbiased reward by learning from the actions sampled from the proposal policy.

Taking inspiration from this design, we formulate our Fat-to-Thin Policy Optimization by maintaining two policies for different purposes: an infinite-support policy (fat) that learns from the offline dataset, parametrized by $\phi$; and a sparse policy (thin) that learns from the actions generated by the fat policy, parametrized by $\theta$. Specifically, we follow the common practice Eq. (1) for the first stage and minimizes the reverse KL divergence between the two policies. The loss functions of FtTPO can be expressed as the following:

$$\mathcal{L}_{\text{Fat, Proposal}}(\phi) := \mathbb{E}_{\substack{s \sim \mathcal{D} \\ a \sim \pi_{\mathcal{D}}}} \left[ -w(s, a) \ln \pi_\phi(a|s) \right], \tag{3}$$

$$\mathcal{L}_{\text{Thin, Actor}}(\theta) := \mathbb{E}_{s \sim \mathcal{D}} \left[ D_{KL}(\pi_\theta(\cdot|s) \,||\, \pi_\phi(\cdot|s)) \right]$$

$$\approx \mathbb{E}_{\substack{s \sim \mathcal{D} \\ a \sim \pi_\theta}} \left[ \frac{\pi_\phi(a|s)}{\pi_\theta(a|s)} + \ln \pi_\theta(a|s) - \ln \pi_\phi(a|s) - 1 \right]. \tag{4}$$

In the proposal loss, we have an additional coefficient $w(s, a)$ for weighting the importance of actions. By letting $w(s, a) = 1$, it is clear that $\mathcal{L}_{\text{Fat, Proposal}}$ recovers the forward KL case in Eq. (1). We discuss more choices that can facilitate learning in Section 4.3. Computing the KL divergence involving a sparse policy can be very unstable. Therefore, we propose to do the following two steps for the actor loss: (1) before every update of the actor, copy the proposal mean to the actor; (2) use an unbiased estimator of KL divergence that has less variance (Schulmann, 2020).

While $\mathcal{L}_{\text{Fat, Proposal}}(\phi)$ and $\mathcal{L}_{\text{Thin, Actor}}(\theta)$ alone – respectively corresponding to forward and reverse KL minimization – cannot learn sparse policies, FtTPO can successfully learn one that competes favorably against popular existing methods. We attribute the success to the two-stage framework where the thin policy allows for further improvement of the behavior-constrained fat policy. It is also worth noting that many other alternatives are possible for learning the fat and thin policies. Our choice is based on simplicity and performance. Some more sophisticated methods like SPOT (Wu et al., 2022) do not lead to better performance. We discuss them in related work and verify that in the experiment section.

## 4.2 INSTANTIATING FtTPO WITH $q$-GAUSSIANS

While in principle any distribution can be used for the fat and thin policy, we consider the $q$-Gaussian family which includes both heavy-tailed and sparse members (Naudts, 2010; Zhu et al., 2024b):

$$\pi_{\mathcal{N}_q}(a|s) = \frac{1}{Z_q(s)} \exp_q \left( -\frac{(a - \mu(s))^2}{2\sigma(s)^2} \right),$$

$$\text{where } Z_q(s) = \begin{cases} \sigma(s) \sqrt{\frac{\pi}{1-q}} \, \Gamma\left(\frac{1}{1-q} + 1\right) / \Gamma\left(\frac{1}{1-q} + \frac{3}{2}\right) & \text{if } -\infty < q < 1, \\ \sigma(s) \sqrt{\frac{\pi}{q-1}} \, \Gamma\left(\frac{1}{q-1} - \frac{1}{2}\right) / \Gamma\left(\frac{1}{q-1}\right) & \text{if } 1 < q < 3. \end{cases} \tag{5}$$

Here, $\mu(s), \sigma(s)$ refer to the state-conditioned mean and standard deviation. $Z_q$ denotes the normalization constant that ensures the policy integrates to 1. Without confusion we omit the dependence on the state. Recall that when $q = 1$ the $q$-Gaussian recovers the standard Gaussian distribution. $q < 1$ corresponds to sparse distributions and $1 < q < 3$ to the heavy-tailed.

We can set $\pi_\phi$ as a heavy-tailed $q$-Gaussian and $\pi_\theta$ as a sparse $q$-Gaussian in Eq. (3). We choose $q = 0$ for the thin policy and $q = 2$ for the fat policy, which are standard values (Chow et al., 2018; Zhu et al., 2024b). To sample from the $q$-Gaussians, we resort to the Generalized Box-Müller Method (GBMM) to map uniform random variables to $q$-Gaussian variables for all $q < 3$ (Thistleton et al.,

2007). Specifically, we sample $u_1, u_2 \sim \texttt{Uniform}(0, 1)$ and compute the following:

$$z_1 = \sqrt{-2 \ln_{q'}(u_1)} \cdot \cos(2\pi u_2), \qquad z_2 = \sqrt{-2 \ln_{q'}(u_1)} \cdot \sin(2\pi u_2), \qquad (6)$$

then each of $z_1, z_2$ is a standard $q$-Gaussian with new index $q = (3q' - 1)/(q' + 1)$. Often we know the desired $q$ in advance (as is the case for our fat and thin policies $q = 2$ and $q = 0$), we simply generate variables by using $q' = (q - 1)/(3 - q)$. To sample from $\mathcal{N}_q(\boldsymbol{\mu}, \Sigma)$ where $\boldsymbol{\mu}$ denotes an $N$-dimensional mean vector and $\Sigma$ an $N \times N$ covariance matrix, we sample uniform random vectors $\boldsymbol{u}_1, \boldsymbol{u}_2 \sim \texttt{Uniform}(0, 1)^N$ and compute the transformed $\boldsymbol{z}$ entry-wise via the GBMM. The desired random vector is given by $\boldsymbol{\mu} + \Sigma^{\frac{1}{2}} \boldsymbol{z}$.

### 4.3 $q$-EXPONENTIAL AS WEIGHTING COEFFICIENT

Many weighting schemes have been proposed to promote "good actions" in Eq. (3). One prominent example is the exponential advantage function $w(s, a) := \exp\left(\frac{Q(s,a) - V(s)}{\tau}\right)$, where $\tau$ is the temperature coefficient (Peng et al., 2020; Nair et al., 2021). When an action has large advantage value, its log-likelihood will be emphasized and the others implicitly de-emphasized. This scheme is the basis of many state-of-the-art algorithms (Kostrikov et al., 2022; Garg et al., 2023; Xu et al., 2023). However, it is worth noting that due to the non-negativity of $\exp$ function, all actions will receive nonzero weights regardless of how bad they are.

We propose to change the exponential function to a $q$-exp: $w(s, a) := \exp_q\left(\frac{Q(s,a) - V(s)}{\tau}\right)$ with $q < 1$. The $q$-exp weights have the desirable effect of filtering out "bad actions" with low advantages thanks to the sparsity of $q$-exp:

$$\mathbb{1}\left\{\left(1 + (1 - q) \cdot \frac{Q(s, a) - V(s)}{\tau}\right)^{\frac{1}{1-q}} \geq 0\right\} \cdot \left(1 + (1 - q) \cdot \frac{Q(s, a) - V(s)}{\tau}\right)^{\frac{1}{1-q}}.$$

Since the root does not affect the sign, it is clear that the $q$-exp weights will truncate actions with advantage $Q(s, a) - V(s) < -\frac{\tau}{1-q}$. But higher $q$ can shrink the magnitude of the advantage function. Since it has been shown by (Zhu et al., 2023) that the role of $q$ and $\tau$ are interchangeable in this regard, we can safely choose $q = 0$. The $q$-exponential advantage weighting has been shown to achieve superior performance in the single actor policy setting (Xu et al., 2023; Zhu et al., 2024a). The experiments verify the sparse policy performs competitive against the original method.

FtTPO is listed in Algorithm 3. For simplicity, we assume at every policy update step $t$ the action value $Q_{\psi_t}$ parametrized by $\psi_t$ and state value $V_{\zeta_t}$ parametrized by $\zeta_t$ are available. They are trained by the standard critic learning procedures which will be detailed in the appendix. Recall that we set the entropic index $q_w = 0$ for the weighting coefficient $w(s, a)$, but in principle any $q_w < 1$ will have the filtering property. We initialize by Alg. 1 and sample from the policies by Alg. 2. The loss functions are empirical expectations $\widehat{\mathbb{E}}_{s,a}$ over the sampled states and actions.

---

**Algorithm 1:** $q$-Gaussian Initialization

**Input:** $q_f > 1$ and $q_s < 1$
Init. $\pi_\phi$ by $\mathcal{N}_{q_f}(\boldsymbol{\mu}_\phi, \Sigma_\phi)$ per Eq. (5)
Init. $\pi_\theta$ by $\mathcal{N}_{q_s}(\boldsymbol{\mu}_\theta, \Sigma_\theta)$
**return** $\pi_\phi, \pi_\theta$

---

**Algorithm 2:** $q$-Gaussian Sampling

**Input:** $q', N, \boldsymbol{\mu}, \Sigma$
sample $\boldsymbol{u}_1, \boldsymbol{u}_2 \sim \texttt{Uniform}(0, 1)^N$
compute $\boldsymbol{z} = \sqrt{-2 \ln_{q'}(\boldsymbol{u}_1)} \cdot \cos(2\pi \boldsymbol{u}_2)$
**return** $\boldsymbol{\mu} + \Sigma^{\frac{1}{2}} \boldsymbol{z}$

---

**Algorithm 3:** Fat-to-Thin Policy Optimization

**Input:** $\mathcal{D}, T, \tau > 0, q_w < 1$
Initialize policies by Alg. 1 ;
**while** $t < T$ **do**
    sample states $s$ from dataset $\mathcal{D}$ ;
    sample actions $a$ from behavior policy $\pi_\mathcal{D}$;
    compute $Q_{\psi_t}(s, a)$ and $V_{\zeta_t}(s)$;
    update $\phi_t$ to $\phi_{t+1}$ by minimizing
    $-\widehat{\mathbb{E}}_{s,a}\left[\exp_{q_w}\left(\frac{Q_{\psi_t}(s,a) - V_{\zeta_t}(s)}{\tau}\right) \ln \pi_{\phi_t}(a|s)\right]$;
    sample $b$ from $\pi_{\theta_t}$ by Alg. 2;
    copy $\boldsymbol{\mu}_{\phi_{t+1}}$ to $\boldsymbol{\mu}_{\theta_t}$ ;
    update $\theta_t$ to $\theta_{t+1}$ by minimizing
    $\widehat{\mathbb{E}}_{s,b}\left[\frac{\pi_{\phi_t}(b|s)}{\pi_{\theta_t}(b|s)} - 1 - \ln \frac{\pi_{\phi_t}(b|s)}{\pi_{\theta_t}(b|s)}\right]$;
**end**

---

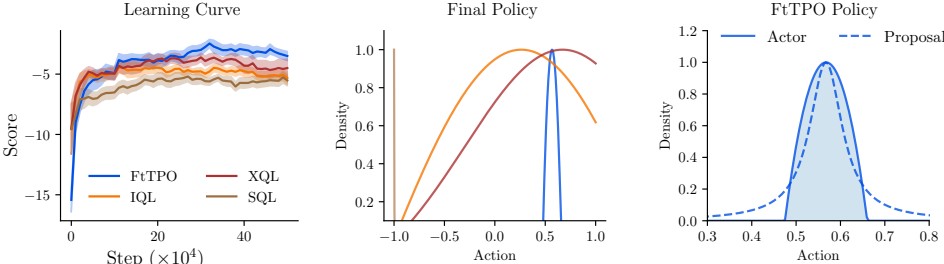

Figure 2: (Left) Scores across the learning. FtTPO attained the highest score, indicating it has learned a safety-aware treatment strategy. The sub-optimal scores of the baselines suggest potential unsafe actions. The solid lines show the mean and the ribbons $95\%$ confidence interval. (Middle) The final policy learned by each algorithm. Only FtTPO managed to learn a sparse yet stochastic policy tightly concentrating around a small band of actions. By contrast, SQL collapsed into a delta-like policy due to approximating a sparse policy with Gaussian. Other baselines have overly large randomness. (Right) The FtTPO actor policy learns from the proposal policy by truncating its heavy tails and retaining only the crucial trunk.

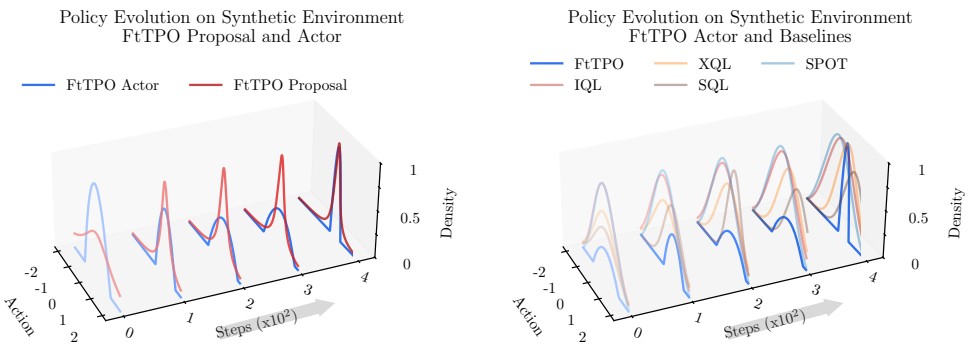

Figure 3: The policy evolution plots over the first 400 updates. LHS shows that the FtTPO proposal policy located a high-reward region and then the actor policy concentrated by removing the heavy tails. RHS compares the FtTPO actor against other baselines using the Squashed Gaussian policy. All baselines tended to have overly large randomness which can result in dangerous dosage.

## 5 EXPERIMENTS

In the experiments, we first verify that FtTPO is capable of learning a sparse and safe policy on a simulated medicine environment. Then on the D4RL Mujoco benchmark, we demonstrate that FtTPO can perform favorably against several popular offline algorithms that by default employ the Gaussian policy. Lastly, we examine in the ablation studies that FtTPO improves on its components. Implementation details are available in Appendix A.

**Baselines.** We choose several state-of-the-art methods as the baselines to verify that FtTPO can competes favorably against them both in in terms of safety and performance. Specifically, we choose XQL (eXtreme Q Learning) that learns a maximum entropy policy without the entropy bias (Garg et al., 2023); SQL (Sparse Q Learning) that approximates an $\alpha$-divergence induced sparse policy with the Gaussian (Xu et al., 2023), InAC (In-sample Softmax Actor-Critic) (Xiao et al., 2023) and IQL (Implicit Q Learning) (Kostrikov et al., 2022) that have been shown to perform very competitively. Implementation details such as the Hyperparameter tuning are provided in Appendix A.2. For the baselines their published settings are used.

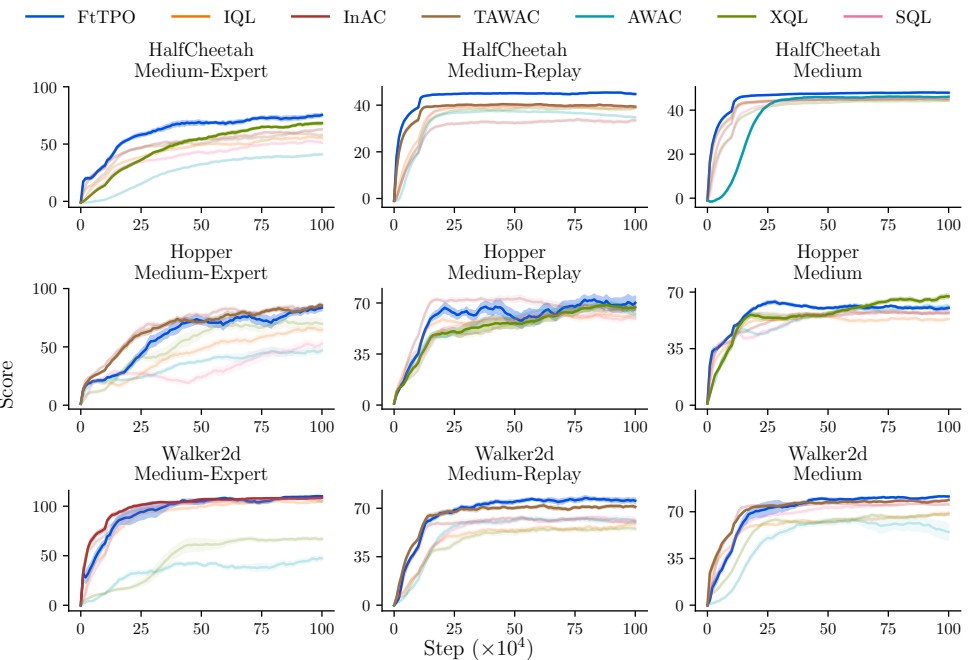

Figure 4: Comparison between FtTPO and the baselines on the MuJoCo. Only FtTPO and the environment-specific best performing algorithm are shown with full transparency. The best algorithm is picked using the final score. All algorithms are run for 10 seeds. The solid lines show the mean and the ribbons 95% confidence interval.

## 5.1 SAFETY-CRITICAL TREATMENT SIMULATION

In this paper we consider the case where safety is explicitly coded into reward, so higher cumulative reward suggests a safety-respecting policy. However, this may not always be true and extra care is required when reward and safety need to be considered separately. We defer such investigation to future study. We opt for the synthetic environment used in (Li et al., 2023) that simulates real circumstances for continuous treatment. Note that only the most challenging environment is used here. The environment has 8-dimensional state space and an unbounded action space. Safety is explicitly coded into the reward function, such that danger is defined as cumulative dosage exceeding a pre-specified threshold. Note that the states are also conditional on past dosage, therefore danger states can vary depending on the dosage in the beginning states, see Appendix A for detail. According to (Li et al., 2023), the optimal dosage is unique, and a high dosage leads to excessive toxicity while a lower dosage is ineffective (Lu et al., 2022). We expect that the sparse policy can achieve superior performance by randomly selecting dosage from a sub-interval and by strictly setting the action probability of other dosage to 0. The offline dataset contains 50 trajectories each comprising 24 steps.

Figure 2 shows the performance. All the algorithms are run for 10 seeds. The solid lines show the mean and the ribbons 95% confidence interval. It is visible that FtTPO outperformed all the baselines in this safety-aware task. All baselines except the SPOT converged to a sub-optimal band of scores as a result of their Squashed Gaussian policies. Indeed, from the middle subplot it can be seen that the baselines tended to have overly large randomness which can sometimes adopt dosage that are not safe. The SQL final policy collapsed as the result of approximating a sparse policy using the Gaussian. By contrast, FtTPO managed to learn a sparse yet stochastic policy tightly concentrating around the optimal action. The last subplot visualizes the synergy in FtTPO: the proposal policy located a high-reward region, and the actor truncated its heavy tails and kept only the essential trunk.

To better visualize the learning process, Figure 3 shows the policy evolution plots for the first 400 updates. The left hand side shows how the FtT proposal and actor policies worked together in locating

a high reward region and then concentrating on it by removing heavy tails. The right hand side compares the FtT actor with the baselines that used the default Squashed Gaussian policy. It is clear that the baselines tended to have overly large randomness spanning the action range $[-2, 2]$ which can result in dangerous dosage.

## 5.2 MuJoCo

The D4RL MuJoCo suite has been a standard benchmark for testing various offline RL algorithms. In this section we compare FtTPO against the baselines on 9 datasets each corresponding to a behavior policy level and environment combination. We also include Tsallis Advantage Weighted Actor-Critic (TAWAC) and AWAC as baselines, which respectively corrrespond to the proposal policy only and the exponential advantage weighting setting. We run all algorithms for $1 \times 10^6$ steps and average over 10 seeds.

Figure 4 shows the full result. Only FtTPO and the environment-specific best algorithm from the baselines are shown with full transparency. Others are shown with low transparency for uncluttered visualization. The best baseline is picked by the final score. Recall that XQL, SQL, InAC, TAWAC are very competent on the Mu-JoCo tasks.

Given that it is commonly perceived that the sparse policies are inherently handicapped at the exploration-exploitation tradeoff, and no past prior work has demonstrated competitive performance to the infinite-support policies, it may come as a surprise that FtTPO performs favorably to or sometimes better than those state-of-the-art algorithms even with a sparse actor. This result suggests that full-support randomness might not be necessary. Rather, learning a sparse policy yet not deterministic, with randomness only in a fixed region could be more preferable. This can be seen from Figure 5 plotting evolution of the 1st action dimension. See Figure 15 for all dimensions.

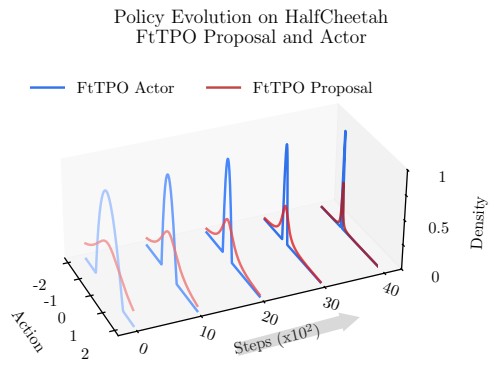

Figure 5: Policy evolution of the 1st action dimension of FtTPO on HalfCheetah Medium-Expert.

## 5.3 Ablation Study

In the ablation studies we are interested in answering the following questions: (1) are more sophisticated methods that sample from the learned policy such as SPOT (see equation 7) superior to the simple FtTPO actor KL minimization? (2) is FtTPO inferior to its proposal only setting, which has been shown to be performant in the MuJoCo benchmark? Note that the proposal-only setting is a competitive baseline that learns a heavy-tailed policy rather than a sparse policy. (3) is the heavy-tailed $q$-Gaussian FtTPO proposal policy better than the Gaussian?

To answer these questions, we evaluate the final scores of the aforementioned combinations and visualize the proportion relative to the FtTPO scores in Figure 6. Here, 100% means exactly the same final score as the FtTPO. It can be seen that despite the simplicity of FtTPO, (1) the KL minimization actor loss is on par with sophisticated SPOT actor learning loss, with FtTPO-SPOT performed slightly better on only 3 environments. (2) even FtTPO outputs a sparse policy, the performance is no worse than the proposal policy which yields a heavy-tailed policy. (3) overall, FtTPO + Squashed Gaussian policy (FtTPO-SG) performed significantly poorer than FtTPO, with exception only on the Medium-Replay Hopper environment.

## 6 Related Works

FtTPO draws a connection to many existing works. But perhaps the most similar work to ours is the Greedy Actor-Critic (GAC) that adopts a two-stage policy learning scheme for learning unbiased

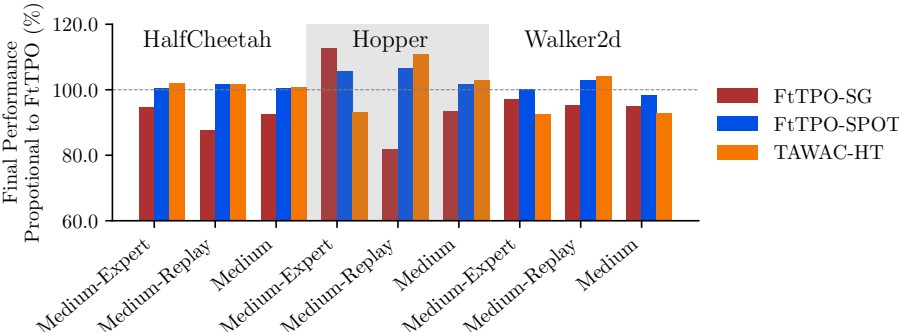

Figure 6: Propotion of final score of FtTPO-SPOT and TAWAC-HT relative to FtTPO. The similar score verifies that the simple FtTPO actor loss is robust and does not lead to inferior performance compared to more complex actor learning procedure (FtTPO-SPOT) and to the FtTPO proposal only setting (TAWAC-HT). The overall better performance to FtTPO-SG (Squashed Gaussian) validates the heavy-tailed proposal policy.

reward-maximizing solutions. There are other methods allowing sampling from thin policy such as the Supported Policy OpTimization (SPOT), which can be used for learning the thin policy. We compare against it in the ablation study. Finally, the $q$-exponential distributions which are frequently used in the physics literature are reviewed.

**Two-stage policy optimization.** Greedy Actor-Critic (GAC) is an online algorithm that considers the conditional cross-entropy method as its actor loss function (Rubinstein, 1999). GAC maintains a entropy-regularized proposal policy to provide high-valued actions for updating the unregularized actor interacting with the environment (Neumann et al., 2023). Specifically, GAC optimizes the following actor losses:

$$\mathcal{L}_{\text{GAC, Proposal}}(\phi) := \mathbb{E}_{\substack{s \sim \mu \\ a \sim I(k, \pi_\phi)}} \left[ -\ln \pi_\phi(a|s) - \alpha \mathcal{H}\left(\pi_\phi(\cdot|s)\right) \right],$$

$$\mathcal{L}_{\text{GAC, Actor}}(\theta) := \mathbb{E}_{\substack{s \sim \mu \\ a \sim I(k, \pi_\phi)}} \left[ -\ln \pi_\theta(a|s) \right],$$

where $\mathcal{H}(\pi)$ denotes the Shannon entropy and $\alpha > 0$ the coefficient. $I(k, \pi_\phi)$ denotes the set of the top $k\%$ actions: the action values of the sampled actions are computed and ranked, and then the actions of the top $k\%$ are extracted. GAC maximizes entropy-regularized log-likelihood for the proposal policy to facilitate exploration and log-likelihood for the actor. We find the set $I(k, \pi_\phi)$ does not bring noticeable improvement. Therefore, we simply use all samples from the policy.

**In-support sampling and updating.** Unlike other methods that sample from the dataset or the behavior policy, Supported Policy OpTimization (SPOT) constrains the policy to be close to the behavior policy by sampling from the learned policy itself (Wu et al., 2022). SPOT learns an actor policy $\pi_\phi$ that maximizes action value and log-likelihood of the behavior policy acting as a regularization term:

$$\mathcal{L}_{\text{SPOT}}(\phi) := \mathbb{E}_{\substack{s \sim \mathcal{D} \\ a \sim \pi_\phi}} \left[ -Q(s, a) - \alpha \ln \pi_\mathcal{D}(a|s) \right], \tag{7}$$

SPOT samples actions from $\pi_\phi$ and imposes constraints directly on the density. While SPOT can also be used for learning the thin policy learning step, we find that empirically it is on better than the simple KL minimization. We show the results in the ablation study Section 5.3.

**Sparse and heavy-tailed policies for RL.** The Gaussian policy has been the default choice for handling continuous action spaces. Some research works explored heavy-tailed policies such as the Student's t (Kobayashi, 2019) or more generally the heavy-tailed $q$-Gaussian (Zhu et al., 2024b). These attempts showed promising performance of the heavy-tailed policies as alternatives to the Gaussian. Li et al. (2023) explored sparse $q$-Gaussian with $q = 0$ implemented by the kernel embedding and fixed-length trajectories. But they did not investigate the out-of-support issue and their implementation required manual design of basis functions specific to simple environments. We provide a general framework empowered by deep networks to learn arbitrary $q$-Gaussians that perform favorably in high-dimensional tasks.

## 7 CONCLUSION

Sparse policies combined with the offline reinforcement learning framework gave rise to a promising new paradigm. It became possible to obtain a safety-aware yet exploratory sparse policy important for realistic systems completely from logged datasets. However, this combination raised a challenge to the existing offline algorithms that require evaluating dataset actions that may fall outside of the sparse policy's support, giving rise to numerical issues and learning failure.

In this paper we proposed Fat-to-Thin Policy Optimization, the first deep offline learning algorithm addressing this issue. We demonstrated that FtTPO was indeed capable of learning a sparse policy that outperformed the popular algorithms in a safety-critical treatment simulation and on the standard MuJoCo control tasks. Policy evolution plots verified that the ability to tightly concentrate around a subset of actions was the key to its superior performance. We conducted ablation studies to verify that the simplicity and sparse policy did not lead to worse performance than its more complex and heavy-tailed counterparts.

Currently, there are few works that study sparse policies in RL. An open question remains that what specific effect the sparse policies bring, e.g. to quantifying the safety-awareness in offline RL or to the exploitation-exploration tradeoff in the online context. This paper focused on offline learning where no exploration is required. But an interesting future direction is to investigate the theoretical implications brought by the sparse policies. For example, it would be interesting to analyze how exploitation capability is gained at the cost of exploration by truncating the heavy tails.

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

# APPENDIX

## A   IMPLEMENTATION DETAILS

### A.1   SAFETY-CRITICAL TREATMENT SIMULATION

We followed (Li et al., 2023) on reproducing this environment, see their Appendix. D.1 for detail. The simulated environment uses the following paradigm to generate data. The action space is unbounded, and actions are sampled uniformly from the range $(-100, 100)$. But the policy can select actions on the entire real line. The transition dynamics follow $s_i^{t+1} \sim \mathcal{N}(\mu_i^{t+1}, \Sigma)$ with $\mu_i^t = [\mu_{i,1}^t, \ldots, \mu_{i,8}^t]$ and $\Sigma$ being a pre-specified covariance matrix. Specifically,

$$\mu_{i,j}^{t+1} = \frac{\exp\left(\frac{a_i^t}{100} + \mu_{i,j}^t\right) - \exp\left(-\left(\frac{a_i^t}{100} + \mu_{i,j}^t\right)\right)}{\exp\left(\frac{a_i^t}{100} + \mu_{i,j}^t\right) + \exp\left(-\left(\frac{a_i^t}{100} + \mu_{i,j}^t\right)\right)}, \quad \text{if } j = 1, 2, 3, 4,$$

$$\mu_{i,j}^{t+1} = \frac{\exp\left(-\frac{a_i^t}{100} + \mu_{i,j}^t\right) - \exp\left(-\left(-\frac{a_i^t}{100} + \mu_{i,j}^t\right)\right)}{\exp\left(-\frac{a_i^t}{100} + \mu_{i,j}^t\right) + \exp\left(-\left(-\frac{a_i^t}{100} + \mu_{i,j}^t\right)\right)}, \quad \text{if } j = 5, 6, 7, 8.$$

The reward function is given by

$$r_i^t = \left(\frac{s_{i,1}^{t+1}}{2}\right)^3 + \left(\frac{s_{i,2}^{t+1}}{2}\right)^3 + s_{i,3}^{t+1} + s_{i,4}^{t+1} + 2\left[\left(\frac{s_{i,5}^{t+1}}{2}\right)^3 + \left(\frac{s_{i,6}^{t+1}}{2}\right)^3\right] + \frac{1}{2}\left(s_{i,7}^{t+1} + s_{i,8}^{t+1}\right).$$

According to (Li et al., 2023), this environment simulates high-dimension state space and a well-separated reward function. The reward function causes the effecet that selecting non-optimal actions will greatly damage the rewards and increases the risk. The environment is tailored to examining whether sparse policies could identify sub-regions and avoids sub-optimal actions which greatly damage the performance.

### A.2   PARAMETER SELECTION

We provide parameter settings of D4RL experiments in Table 1 and the synthetic environment in Table 2. The environment-specific best hyperparameters are listed in Table 3 and 4, respectively.

In terms of computation time, training FtTPO on average costed around 15 hours for 1 million steps. By contrast, among the baselines InAC took 8.5 hours, IQL 6 hours and TAWAC 6.5 hours. This confirms that by maintaining two policy networks FtTPO costs around double computation time. When interacting with the environment, we sample actions only from the actor policy, which costs similar time to the Gaussian.

## B   FURTHER RESULTS

Figure 7 visualizes how the sparse policy of FtTPO provides a principled solution to ensuring safety. The left figure compares FtTPO actor against IQL + Gaussian on the HalfCheetah. It is visible that by clipping the policy, IQL Gaussian causes excessive density falling on the clipping boundary due to normalization constraint. As a result, naive clipping can cause the policy to be a Bang-bang controller (Seyde et al., 2021) that picks actions almost exclusively on the boundary, leading to unsafe actions. The right figure shows the FtTPO actor and reward curves (gray) along with policy evolution. By the definition of reward function, it depends recursively on the past actions. Therefore, a dangerous action can lead to potential future low reward. It is crucial that the agent is capable of utilizing not overly large yet effective dosage at each step to ensure safety. Since the low point of rewards does not reach zero, FtTPO does not incur danger during the evolution.

Figure 8 shows the learning curves of ablation studies. The first shows FtTPO versus FtTPO-SPOT vs FtTPO-SG and the second FtTPO versus TAWAC-HT. Recall that FtTPO-SPOT replaces the actor with SPOT for learning a sparse policy. FtTPO-SG refers to replacing the proposal policy to the

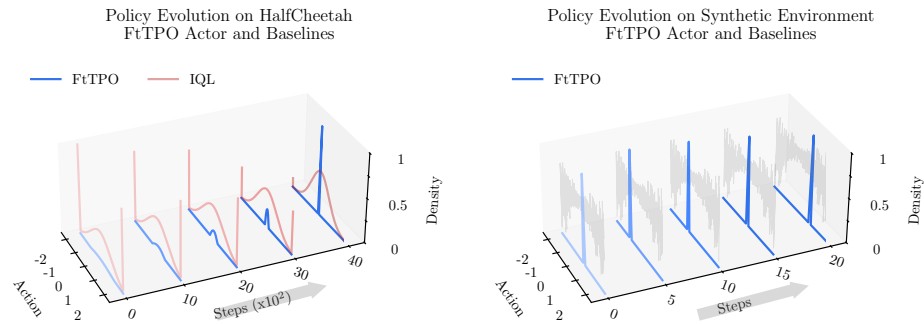

Figure 7: (Left) FtTPO actor compares against IQL + Gaussian on the HalfCheetah. It is visible that by clipping the Gaussian policy, IQL Gaussian causes excessive density falling on the clipping boundary due to normalization constraint. This can lead to choosing actions almost exclusively from the boundary and hence unsafe actions. (Right) FtTPO actor reward curves (gray) along with policy evolution. Since the low point of rewards does not reach zero, by the definition of reward function FtTPO does not incur danger during the evolution.

| Parameter | Value |
|---|---|
| Learning rate | **FtT**: Swept in $\{1e-3, 3e-4\}$ 
 **Baselines**: Swept in $\{3 \times 10^{-3}, 1 \times 10^{-3}, 3 \times 10^{-4}, 1 \times 10^{-4}\}$ |
| Weights | **FtT**: Swept in $\{1.0, 0.5, 0.01\}$ 
 **Baselines**: Same as the number reported in 
 the publication of each algorithm. 
 Except in TAWAC + medium datasets, the value was 
 swept in $\{1.0, 0.5, 0.01\}$. |
| Discount rate | 0.99 |
| Timeout | 1000 |
| Training Iterations | 1,000,000 |
| Hidden size of Value network | 256 |
| Hidden layers of Value network | 2 |
| Hidden size of Policy network | 256 |
| Hidden layers of Policy network | 2 |
| Minibatch size | 256 |
| Adam.$\beta_1$ | 0.9 |
| Adam.$\beta_2$ | 0.99 |
| Target network synchronization | Polyak averaging with $\alpha = 0.005$ |
| Number of seeds for sweeping | 5 |
| Number of seeds for the best setting | 10 |
| STD in sparse policy | Clipped at the upper bound of the action space |

Table 1: Default parameters and sweeping choices in D4RL.

Squashed Gaussian policy. TAWAC-HT is TAWAC equipped with the heavy-tailed $q$-Gaussian, which is the proposal policy only setting.

The bar plot Figure 6 was made using the final scores. But it is clear from the curves that FtTPO is similar to FtTPO-SPOT also in the sense of area under curve (AUC), and they are both slightly better than FtTPO-SG overall. For the lower plot, it is clear that the sparse policy of FtTPO leads to roughly the same performance as the heavy-tailed policy. Therefore, performance is not a concern when replacing the Gaussian/heavy-tailed policies with the sparse one.

We also visualized the policy evolution in HalfCheetah medium-expert dataset in Figure 15. It can be seen that the same trend

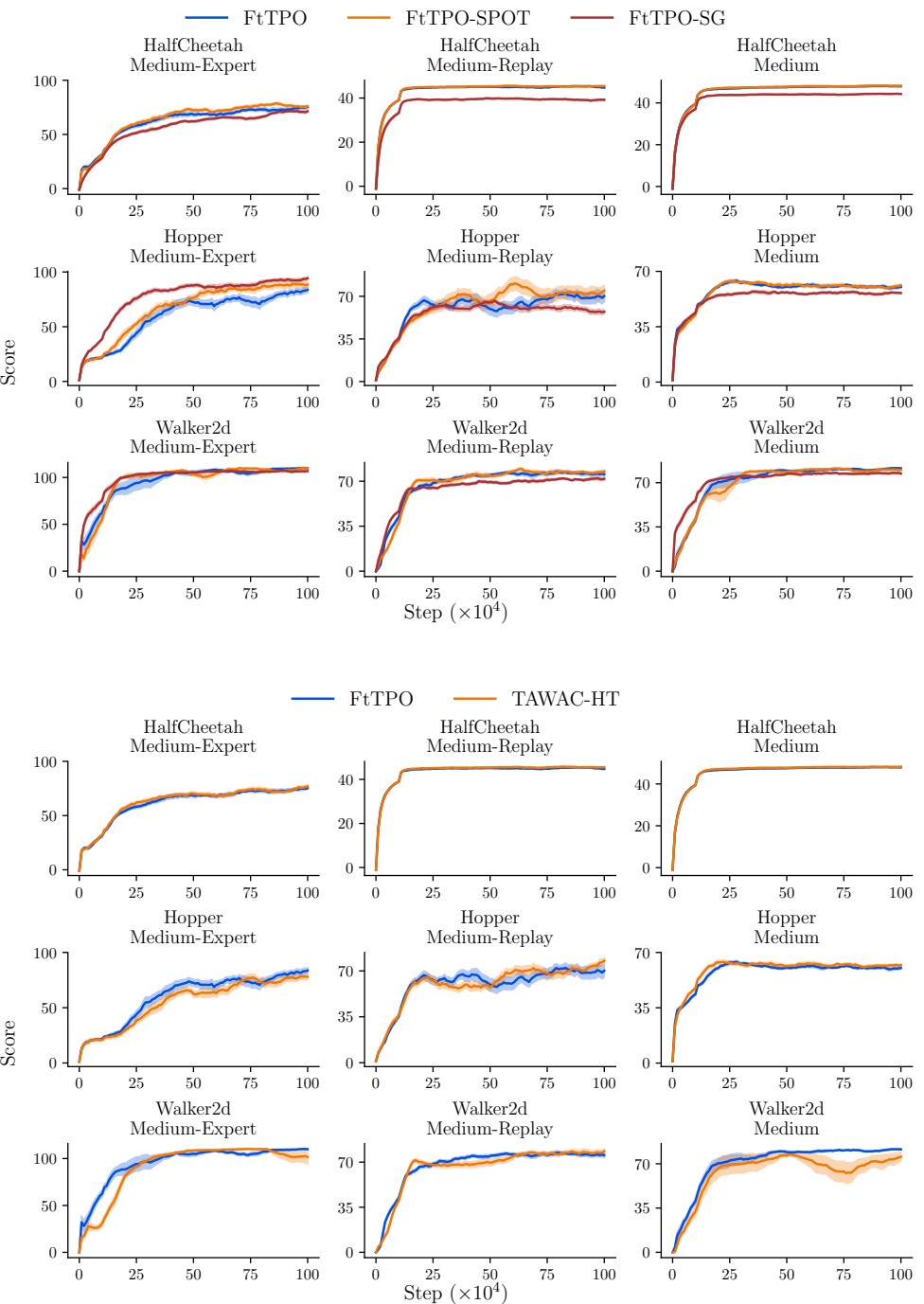

Figure 8: (Upper) Learning curves of the ablation study FtTPO against FtTPO-SPOT and FtTPO-SG (Squashed Gaussian). (Lower) Learning curves of the ablation study FtTPO against the proposal only setting TAWAC-HT (Tsallis Advantage Weighted Actor-Critic-Heavy-Tailed).

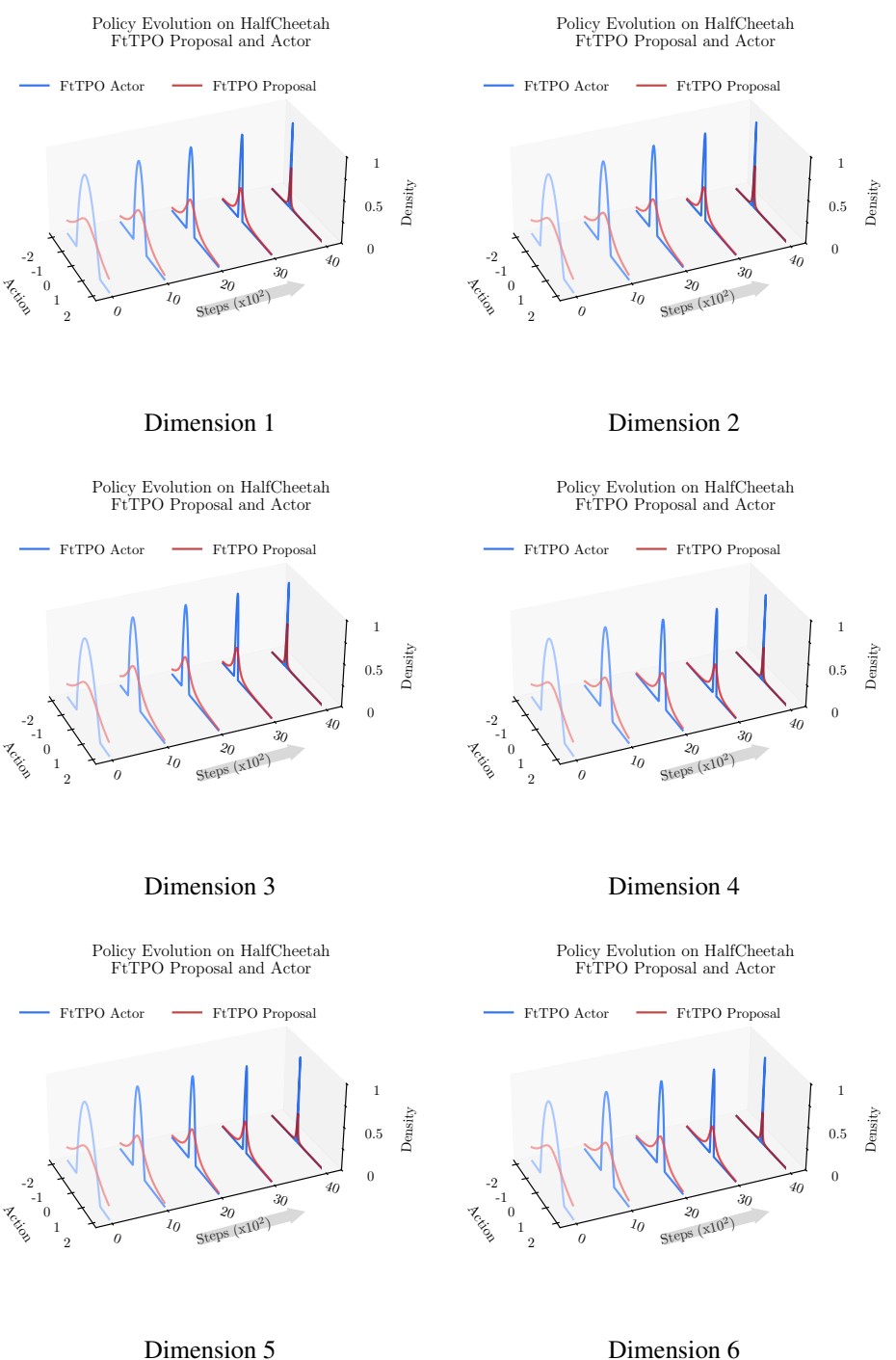

Figure 15: The policy evolution plots over the first 4000 updates on HalfCheetah medium-expert dataset. The FtTPO actor policy concentrated by removing the heavy tails.

| Parameter | Value |
|---|---|
| Learning rate | **FtT**: Swept in $\{3 \times 10^{-4}, 1 \times 10^{-4}, 3 \times 10^{-5}\}$
**Baselines**: Swept in $\{3 \times 10^{-3}, 1 \times 10^{-3}, 3 \times 10^{-4}, 1 \times 10^{-4}\}$ |
| Weights | **FtT**: Swept in $\{1.0, 0.5, 0.1, 0.01\}$
**IQL**: temperature: $\{1/3, 0.7\}$; expectile: $\{0.7, 0.8, 0.9\}$
**SQL, XQL**: $\{2, 5\}$ |
| Discount rate | 0.9 |
| Timeout | 24 |
| Training Iterations | 500,000 |
| Hidden size of Value network | 256 |
| Hidden layers of Value network | 2 |
| Hidden size of Policy network | 256 |
| Hidden layers of Policy network | 2 |
| Minibatch size | 256 |
| Adam.$\beta_1$ | 0.9 |
| Adam.$\beta_2$ | 0.99 |
| Target network synchronization | Polyak averaging with $\alpha = 0.005$ |
| Number of seeds for sweeping | 5 |
| Number of seeds for the best setting | 10 |
| STD in sparse policy | Clipped at the upper bound of the action space |

Table 2: Default parameters and sweeping choices in Synthetic environment.

| Dataset | FtTPO | FtTPO-SG | FtTPO-SPOT |
|---|---|---|---|
| HalfCheetah-Medium-Expert | 0.001 | 0.001 | 0.001 |
| HalfCheetah-Medium-Replay | 0.001 | 0.001 | 0.001 |
| HalfCheetah-Medium | 0.001 | 0.001 | 0.001 |
| Hopper-Medium-Expert | 0.001 | 0.001 | 0.001 |
| Hopper-Medium-Replay | 0.001 | 0.0003 | 0.0003 |
| Hopper-Medium | 0.001 | 0.001 | 0.001 |
| Walker2d-Medium-Expert | 0.001 | 0.001 | 0.0003 |
| Walker2d-Medium-Replay | 0.001 | 0.001 | 0.0003 |
| Walker2d-Medium | 0.001 | 0.001 | 0.001 |
| SimEnv3-Random | 0.0003 | 3e-05 | 3e-05 |

Table 3: Best $\tau$ for FtTPO and variants.

| Dataset | FtTPO | FtTPO-SG | FtTPO-SPOT |
|---|---|---|---|
| HalfCheetah-Medium-Expert | 1.0 | 1.0 | 1.0 |
| HalfCheetah-Medium-Replay | 0.5 | 0.5 | 0.01 |
| HalfCheetah-Medium | 0.01 | 0.01 | 0.01 |
| Hopper-Medium-Expert | 1.0 | 1.0 | 1.0 |
| Hopper-Medium-Replay | 1.0 | 0.5 | 0.01 |
| Hopper-Medium | 0.01 | 0.01 | 0.01 |
| Walker2d-Medium-Expert | 0.5 | 1.0 | 1.0 |
| Walker2d-Medium-Replay | 0.01 | 1.0 | 0.01 |
| Walker2d-Medium | 1.0 | 1.0 | 0.5 |
| SimEnv3-Random | 0.01 | 0.5 | 0.5 |

Table 4: Best temperatures for FtTPO and variants.

