# OpenReview forum: "Fat-to-Thin Policy Optimization: Offline Reinforcement Learning with Sparse Policies"
_ICLR.cc/2025/Conference — ICLR 2025 Poster_

### Official Review · Reviewer_jcMU · 2024-11-02

**Soundness:** 3
**Presentation:** 4
**Contribution:** 3
**Rating:** 8
**Confidence:** 3

**Summary:**

This paper studies offline reinforcement learning in continuous action settings where it is desired to learn a sparse policy (that has zero probability for some actions) that is stochastic. This occurs, for instance, when there is a safety aspect and so some parts of the action space should have zero probability. This paper creates a new algorithm for this setting based on a two-stage actor-critic approach, where a non-sparse proposal policy learns and injects knowledge into a sparse actor. There are quite a few design decisions along the way. The resulting method FtTPO is compared to a large number of existing methods on both safety-motivated and general RL (Mujoco) benchmarks. On the safety benchmarks, it is the strongest, and on general RL it is very competitive.

**Strengths:**

- The paper is excellent at contextualizing its contributions relative to existing work.
- The description of the algorithm, the design decisions, and the motivations behind the design decisions is clear.
- FtTPO makes design decisions that are different than previous algorithms.
- Experimental results are strong against a wide variety of relevant benchmarks. There is enough task variation.
- The demonstration that there are sparse, strong, policies for Mujoco tasks is interesting and provocative.
- Hyperparameter selection is clearly explained.

**Weaknesses:**

- The paper is driven purely by intuition and empirical results. There are several cases where there is not a clear understanding of why a certain approach does not work (e.g., reversing the direction of the KL divergence).
- There is no comparison of computational cost (time) of the different methods.

**Questions:**

1. For 5.1, how many seeds where used?
2. What is shown by the shaded area on all graphs?
3. How does hyperparameter tuning FtTPO compare to the other methods? (I see Tables 1 and 2—I'm interested in a qualitative statement).
4. How do these methods compare in terms of computational time?

---

> ### Author Response · Authors · 2024-11-20
>
> We thank the reviewer for the valuable feedback and appreciation of our work.
>
> We agree that FtTPO is indeed somewhat heuristic-driven, we would like to point out that our design choice was based on the observation that neither forward KL nor reverse KL was capable of achieving the goal. For the reason of failure of reverse KL, as we conjectured in Section 3.2,  it could be due to that the support is finite, the sampled actions may concentrate around its mode and therefore no significant update can be performed, resulting in extremely slow learning. This is further compounded by the fact that the standard deviation tends to become smaller as a result of optimization, so the samples $a\sim\pi_\phi$ will cover an increasingly shrunk region.
>
> Regarding computational considerations, we have appended the draft with the following details:
> 1. we have added that 10 seeds are used for Section 5.1 and 5.2
> 2. we have added explanation to captions of Figure 2 and 4 that the shaded area indicates 95% confidence region
> 3. FtTPO requires sweeping the temperature $\tau$ in addition to the learning rate. Therefore, the number of hyperparameters is the same with popular algorithms like AWAC or IQL.
> 4. The training for FtTPO on average took around 15 hours for 1 million steps. By contrast, among the baselines InAC took 8.5 hours,IQL 6 hours and TAWAC 6.5 hours. This confirms that by maintaining two policy networks FtTPO costs around double computation time. When interacting with the environment, we sample actions only from the actor policy, so the time taken is similar to a standard algorithm. The explanation has been added to Appendix A.2

---

### Official Review · Reviewer_NgxJ · 2024-11-02

**Soundness:** 1
**Presentation:** 3
**Contribution:** 2
**Rating:** 3
**Confidence:** 4

**Summary:**

This paper proposes a novel architecture for learning sparse policies in an offline RL paradigm. Specifically, the authors address issues of domain alignment when training sparse policies, whereby the sparse policy may not support actions within the offline dataset. The induced sparse policy placing measure 0 probability over part of the offline dataset can result in numerical instabilities, for example, when using a log likelihood loss.
The proposed architecture requires first learning a “fat” (or non-sparse policy) policy i.e., a policy with at least the support of the offline dataset (the authors propose a Gaussian), which is subsequently refined to a “thin” policy using existing methods for learning sparse policies i.e., using the reverse KL divergence.

**Strengths:**

-	Originality: To the best of my knowledge the method of induing a sparse policy from one with a larger support has not been explored and as such, the research is original.
-	Clarity: The paper is well presented and written in the sense that relevant (to the best of my knowledge) existing work is contextualised, the proposed solution is well defined and the paper is generally easy to follow.
-	Quality and Significance: The problem is well motivated and thus assuming the solution can be demonstrated to provide significant performance improvements over existing “ad-hoc” approaches, the “fat to thin” architecture  would be of great significance to practioners aiming to learn sparse policies in an offline setting. Furthermore, the breath of evaluation benchmarks used is reasonable in the sense that a reasonable number of challenging offline RL benchmarks have been used.

**Weaknesses:**

The core weakness of the paper is the experimental approach. The authors consistently draw conclusions that it is “surprising” that the proposed architecture performs so well given the “sparse policy” (referenced in section 5.2) or given the relative simplicity of the proposed architecture (referenced in section 5.3). I disagree with this line of analysis for several reasons:
-	A sparse policy working “surprisingly well” is not a substantively validated claim as the authors do not explain why it is surprising. Without this additional analysis, the performance of the proposed method is inconsistent and only obtains notable performance on the Medium-Expert and Medium-Replay HalfCheetah datasets.
-	The authors have not made clear why the proposed architecture is less complex and I would contest, that given two policies are required to be maintained, the proposed architecture is actually more complex than the existing baseline approaches.
Broadly speaking, the proposed architecture appears to be overly complex and the experimental direction of demonstrating that the architecture performs “at least as well as” existing methods is not strong enough to support this additional complexity.
Additionally, it is not completely convincing that the proposed architecture is necessary. Based on Figure 6, the proposal only policy (utilising existing and simpler methods for learning sparse policies) performs on par with the proposed architecture. Furthermore, from Figure 5, the proposal policy appears to already learn a relatively concentrated policy. As such, it is unclear why using only proposal policy and preventing actions being selected outside X standard deviations would not achieve the desired result (assuming an improper distribution of the resulting policy is reasonable).
Overall, I would encourage the authors to revisit the experiments and strive to obtain stronger results that demonstrate solid benefits of the algorithm. Deriving a novel architecture and having it reasonably converge is no mean feat however, the results presented in the paper are not yet ready for publishing. It might be worth exploring why the proposed architecture performed better on the aforementioned HalfCheetah environments to understand the types of environments where the fat-to-thin model is strongly beneficial.

**Questions:**

-	Understanding why the authors feel the fat-to-thin architecture is less complex than existing methods and why it is surprising the a sparse policy competes with existing methods would help in changing my opinion.

---

> ### Author Response · Authors · 2024-11-20
> **clarifying a potential misconception**
>
> We thank the reviewer for the helpful feedback. We would like to explain our method in more detail and point out a potential misconception about our paper.
>
> **Regarding why the results are surprising**:
>
> First we would like to point out that the paper aims to learn a sparse policy which has not been possible in the deep offline RL context before, due to the out-of-support issue incurred by the sparse policies. Therefore, the utility and performance of sparse policies have never been examined in the literature. It is commonly believed that sparse policies are not performant because they are inherently handicapped at the exploration-exploitation tradeoff due to the sparsity [Lee 2018, Nachum 2018, Zhu 2024]. Our work for the first time provides a framework to learn sparse policies in the offline context, and verifies that they can be as performant as Gaussian/heavy-tailed policies, not only on specific safety-critical tasks but also on the standard offline RL benchmarks where the Gaussian dominates. Therefore, showing sparse policies are at least as well as existing methods that use Gaussian/heavy-tailed is important, because our method opens up a possibility of utilizing sparse policies not only for safety-critical tasks where they fit more than infinite-support policies, but also for general environments where they can be as good as the Gaussian.
>
> We thank the reviewer’s questions regarding that the reader may not fully appreciate why the result is surprising. We have correspondingly modified our text to help the reader better understand the contribution of the paper. Specifically, we added the following  to the introduction:
>
> > As it is commonly perceived that the sparse policies are inherently handicapped at the
> exploration-exploitation tradeoff, we find it surprising that the sparse policy learned by FtTPO can outperform full-support policies.
>
> and the following to Section 5.2:
>
> > Given that it is commonly perceived that the sparse policies are inherently handicapped at the exploration-exploitation tradeoff, and no past prior work has demonstrated competitive performance to the infinite-support policies, it may come as a surprise that FtTPO performs favorably to or sometimes better than those state-of-the-art algorithms even with a sparse actor.
>
> **Regarding the misconception and why the additional complexity on architecture is necessary**:
>
> We would like to point out a potential misconception of the reviewer: *the proposal-only setting learns a heavy-tailed distribution rather than a sparse policy*. This setting is actually a very strong baseline [Zhu et al. 2024] that leverages the properties of heavy-tailed distributions [Kobayashi, 2019, Zhu et al., 2024]. It cannot be used directly for learning a sparse policy due to the reason stated for Eq. (1): the log-likelihood will have undefined values for out-of-support actions. Therefore, by comparing FtTPO against the proposal-only setting, we are actually comparing a sparse policy against a known strong baseline utilizing a heavy tailed policy. Hence, we believe that being able to be on par with such baselines on standard RL problems proves the value of our method.
>
> We acknowledge that the proposal-only setting was not very well explained in the paper, we have added the following explanatory text to Section 5.3:
>
> > Note that the proposal-only setting is a competitive baseline that learns a heavy-tailed policy rather than a sparse policy.
>
>
> **Regarding why not just heavy-tailed or clipping X std from the mean**:
>
> Regarding concentration of the proposal policy, by examining the rightmost of Figure 2 and Figure 15, it is visible that the heavy-tailed policy can differ considerably from the sparse policy, as they have heavy tails decaying slowly and may contribute to selecting unsafe actions.
>
> By clipping a policy X standard deviations from the mean, we will have a biased policy with significant densities on the border of clipping, which can be considerably different from the intended policy shape. To verify this phenomenon, we have included a new Figure 7, whose LHS compares the IQL + Gaussian to the proposed method. Because on this environment the action space is bounded, density outside the range is clipped and the resulting policy is very different from FtTPO.
>
> We have added the following explanatory text to Appendix B:
>
> > Figure 7 visualizes how the sparse policy of FtTPO provides a principled solution to ensuring safety.
> The left figure compares FtTPO actor against IQL + Gaussian on the HalfCheetah. It is visible that
> by clipping the policy, IQL Gaussian causes excessive density falling on the clipping boundary due to
> normalization constraint. As a result, naive clipping can cause the policy to be a Bang-bang controller
> (Seyde et al., 2021) that picks actions almost exclusively on the boundary, leading to unsafe actions.

---

> ### Author Response · Authors · 2024-11-24
> **addressing additional questions**
>
> Dear Reviewer,
>
> We would like to thank you again for your thoughtful feedback. As the deadline of the discussion phase is approaching, we would like to kindly ask if our rebuttal has addressed your concerns, especially regarding the novelty of sparse policies, level of performance and additional complexity. We value the opportunity to engage with you during this rebuttal phase, as your insights are instrumental in improving the quality of our work. Please do not hesitate to let us know if you have further questions. Thank you for your support and looking forward to hearing from you.
>
> Sincerely,\
> The Authors

---

### Official Review · Reviewer_fP7N · 2024-11-03

**Soundness:** 3
**Presentation:** 3
**Contribution:** 3
**Rating:** 8
**Confidence:** 3

**Summary:**

This paper proposes a novel approach for offline reinforcement learning with sparse policies. Sparse policies have important real-world implications but pose challenges to existing algorithms. The paper presents Fat-to-Thin Policy Optimization (FtTPO), a two-stage learning method that addresses these challenges.

**Strengths:**

The paper presents a novel FtTPO algorithm for handling sparse policies in offline reinforcement learning. This is a new combination of ideas as it builds on two-stage actor-critic methods and uses a fat (infinite-support) policy to inform a thin (sparse) policy. It addresses the previously less-studied problem of out-of-support actions in offline learning with sparse policies, providing a solution where prior works relied on ad hoc methods.

The paper is well-structured, with clear sections for introduction, background, method description, experiments, and related works.

**Weaknesses:**

The safety benefit of having a thin (sparse) policy isn't directly clear to readers. The authors somehow use 'performance' and 'safety' interchangeably and imply that a higher reward means higher safety.  A very common misunderstanding is that 'dangerous action' means 'higher dosage'. For example, the daily insulin dosage for T1 diabetic patients is around 0.5-1 unit per day. A 0.001 unit dosage can be considered a low dosage. However, if such a dosage is given per 2 min, the accumulative dosage will far exceed body tolerance and cause serious damage to the patient. In fact, 'dangerous states and actions' are precise in medicine. In domains such as dynamic treatment regimes, dangerous states are explicitly defined. I encourage the authors to define safety clearly before using it. Besides, the authors can visualize the occurrence of dangerous states following their policy versus baseline policies to see if safety is indeed enhanced.

**Questions:**

NA

---

> ### Author Response · Authors · 2024-11-20
> **problem setting redefined and new figures added**
>
> We sincerely appreciate the reviewer’s valuable suggestion to help better position the paper. Indeed in the paper we used the words safety and performance interchangeably with an assumption that safety is coded into reward explicitly. This is the case for our simulated treatment environment as its reward recursively conditions on past actions. We agree that this may not always be true and we have modified the text accordingly to clarify this point. The beginning paragraph in Section 5.1 now writes:
>
> > **In this paper we consider the case where safety is explicitly coded into reward, so higher cumulative reward suggests a safety-respecting policy. However, this may not always be true and extra care is required when reward and safety need to be considered separately. We defer such investigation to future study**. We opt for the synthetic environment used in (Li et al., 2023) that simulates real circumstances for continuous treatment. Note that only the most challenging environment is used here. The environment has 8-dimensional state space and an unbounded action space. Safety is explicitly coded into the reward function, **such that danger is defined as cumulative dosage exceeding a pre-specified threshold. Note that the states are also conditional on past dosage, therefore danger states can vary depending on the dosage in the beginning states**, see Appendix A for detail. According to (Li et al., 2023), the optimal dosage is unique, and a high dosage leads to excessive toxicity while a lower dosage is ineffective.
>
> Regarding the additional graphs for validating FtTPO. We have included a new Figure 7 (RHS) that helps explain why the safety is ensured by the proposed method.
>
> > The right figure shows the FtTPO actor and reward curves (gray) along with policy evolution. By
> the definition of reward function, it depends recursively on the past actions. Therefore, a dangerous
> action can lead to potential future low reward. It is crucial that the agent is capable of utilizing not
> overly large yet effective dosage at each step to ensure safety. Since the low point of rewards does
> not reach zero, FtTPO does not incur danger during the evolution.
>
> We noted that it is not possible to directly compare all policies on one state because the state-action space is continuous, and they do not experience an exactly same state that allows the direct comparison.

---

### Official Review · Reviewer_vniq · 2024-11-05

**Soundness:** 3
**Presentation:** 4
**Contribution:** 3
**Rating:** 8
**Confidence:** 3

**Summary:**

This paper introduces a novel approach for learning sparse policies within the context of offline RL. Sparse policies are crucial for enhancing safety, as they prevent the policies from considering all possible actions, which can reduce potential risks. Unlike existing methods that rely on ad hoc techniques such as reverse KL divergence or random action replacement, this study proposes the Fat-to-Thin Policy Optimization (FtTPO) algorithm. This algorithm leverages the deformed q-exponential function to parameterize policies and employs a greedy two-stage actor-critic optimization approach. The result is a method that achieves desirable sparsity and outperforms existing techniques in simulated deep offline RL tasks.

**Strengths:**

- The paper presents an innovate approach for learning sparse policies from logged datasets that outperform current offline RL methods.
- The work is empirical rigorous, with sufficient analysis and abalations of the proposed method on a variety of simulated tasks - including safety-critical  bencahmark and Mujoco.
- The paper is clearly written and accessible, making it easy to understand the authors' arguments and methodology.

**Weaknesses:**

- It could be great if the authors can provide insight into why the combination of forward and reverse KL (with the two-stage optimization framework) helps in the first place?
- Another element that the work is missing is to compare how the proposed method works in low-data regime. In a lot of safety-critical settings, size of the datasets are quite limited, so it would have been nice to have that analysis in the paper.

**Questions:**

- A primary factor contributing to the success of the proposed algorithm appears to be the use of the weighted q-exponential function in the objective. According to line 241, $ q = 0 $ is used in practice. If so, then how does that affect filtering of "bad actions"?

- In algorithm 3, won't copying $\mu_{\phi_t}$ to $\mu_{\theta_t}$ violate the original choice of the policy parametrization? Is the sampling procedure still valid after copying only parameters and not changing the associated sampling parameters?


Minor comments:
- How to pronounce FtTPO?
- Typo in line 52 (algorithmsk)

---

> ### Author Response · Authors · 2024-11-20
>
> We thank the reviewer for appreciating our work and providing helpful feedback. We address the reviewer's questions as follows.
>
> **Regarding insights for why the combination of forward and backward KL helps**:\
> We attribute the success of FtTPO to the way two policies are learned: while forward KL cannot be used for learning sparse policy, we can first learn a high-quality heavy-tailed policy from data. Its location and scale provide important knowledge to the sparse policy. This greatly eases the difficulty of learning a sparse policy, whose sparse tails can significantly hinder learning efficiency. This point was briefly discussed in Section 3.2. We have modified it to the following:
>
> > We conjecture that the inability of reverse KL alone is due to the finite support of sparse policies, which cause the “raw” sampled actions to concentrate around its mode and therefore no significant update can be performed, resulting in extremely slow learning.
>
> **Regarding application to the low-data regime**:\
> We thank the reviewer for pointing out potential experiments in the low-data regime. Our simulated treatment experiment has only 50 trajectories and each trajectory contains 24 steps. We believe FtTPO has the potential for some low-data applications. We greatly appreciate the suggestion and plan to do such experiments with more realistic data in the future.
>
> **How $q$ is used to filter out bad actions**:\
> According to the definition in Section 4.3, the weight
> $$ \exp_q\left(\frac{Q(s,a) - V(s)}{\tau}\right) = \mathbb{1}\left[\left(1 + (1 − q) \frac{Q(s,a) - V(s)}{\tau} \right)^\frac{1}{1-q} ≥ 0 \right] · \left(1 + (1 − q) \frac{Q(s,a) - V(s)}{\tau} \right)^\frac{1}{1-q}  $$
> i.e. bad actions are those with advantage values $Q(s,a) - V(s) <  -\frac{\tau}{1-q}$ that cause the term vanish. We see the threshold is controlled by $q$ and $\tau$. When using $q=0$, the threshold is controlled by $\tau$ alone.
>
> **Regarding copying of $\mu_{\phi}$**:\
> Copying the proposal mean to the actor only serves the purpose of facilitating actor learning, since the actor mean and standard deviation are still subject to optimization of the actor and therefore they can change. Since the optimization is performed on the actions sampled from the actor policy, it does not violate its policy parametrization.
>
>
> **Minor comments**:\
> We thank the reviewer for pointing out the typo and have corrected it. We pronounce our method as F-T-T-P-O.

---

### Meta-Review · Area_Chair_tbAe · 2024-12-21

**Metareview:**

This paper studies offline reinforcement learning with sparse policies, addressing the novel and critical issue of combining safety-aware sparse policy learning with logged datasets. It introduces the Fat-to-Thin Policy Optimization (FtTPO) algorithm, leveraging a two-stage actor-critic framework and heavy-tailed proposal policies to enable sparse policy optimization, with strong empirical results on safety-critical and standard benchmarks. The reviewers found the methodological contributions and experiments to be robust and well-motivated, particularly the innovative use of sparse policies in RL. While one reviewer questioned the need for large-scale experiments and perceived complexity, the authors effectively clarified the theoretical and practical relevance of their approach during the rebuttal phase. Overall, the paper is a valuable contribution to reinforcement learning theory and practice.

**Additional Comments On Reviewer Discussion:**

During the rebuttal phase, key points raised by reviewers included the complexity and necessity of the proposed Fat-to-Thin Policy Optimization (FtTPO) architecture, the lack of clarity on why sparse policies are competitive with full-support policies, and concerns about the experimental setup and its sufficiency to validate the method's contributions. Reviewer NgxJ questioned the added complexity of maintaining two policies and suggested simpler baselines, while also challenging the surprising performance of sparse policies. Reviewer jcMU requested clarification on computational overhead and hyperparameter tuning. Reviewer fP7N emphasized the need for clearer articulation of the safety-performance tradeoff and further experimental insights. The authors addressed these points by elaborating on the novelty of sparse policies, clarifying their necessity for safety-critical tasks, adding experimental details (e.g., computational time and hyperparameter settings), and refining explanations of the method's surprising results. After considering the authors' clarifications and revisions, I weighed the theoretical contribution, empirical support, and rebuttal strength, concluding that the paper is acceptable despite some lingering concerns about computational complexity.

---

### Decision · Program_Chairs · 2025-01-22

Accept (Poster)